# Incorporating First Nations, Inuit and Métis Traditional Healing Spaces within a Hospital Context: A Place-Based Study of Three Unique Spaces within Canada’s Oldest and Largest Mental Health Hospital

**DOI:** 10.3390/ijerph21030282

**Published:** 2024-02-28

**Authors:** Vanessa Nadia Ambtman-Smith, Allison Crawford, Jeff D’Hondt, Walter Lindstone, Renee Linklater, Diane Longboat, Chantelle Richmond

**Affiliations:** 1Department of Geography & Environment, Faculty of Social Sciences, Western University, London, ON N6A 3K7E, Canada; chantelle.richmond@uwo.ca; 2Centre for Addiction and Mental Health (CAMH), Toronto, ON M6J 1H4, Canada; allison.crawford@camh.ca (A.C.); jeff.dhondt@camh.ca (J.D.); walter.lindstone@camh.ca (W.L.); renee.linklater@camh.ca (R.L.); diane.longboat@camh.ca (D.L.)

**Keywords:** traditional healing spaces, traditional healing, hospital, healthcare, environmental repossession, Indigenous geographies of health, mental health, sweatlodge, reconciliation

## Abstract

Globally and historically, Indigenous healthcare is efficacious, being rooted in Traditional Healing (TH) practices derived from cosmology and place-based knowledge and practiced on the land. Across Turtle Island, processes of environmental dispossession and colonial oppression have replaced TH practices with a colonial, hospital-based system found to cause added harm to Indigenous Peoples. Growing Indigenous health inequities are compounded by a mental health crisis, which begs reform of healthcare institutions. The implementation of Indigenous knowledge systems in hospital environments has been validated as a critical source of healing for Indigenous patients and communities, prompting many hospitals in Canada to create Traditional Healing Spaces (THSs). After ten years, however, there has been no evaluation of the effectiveness of THSs in Canadian hospitals in supporting healing among Indigenous Peoples. In this paper, our team describes THSs within the Center for Addiction and Mental Health (CAMH), Canada’s oldest and largest mental health hospital. Analyses of 22 interviews with hospital staff and physicians describe CAMH’s THSs, including what they look like, how they are used, and by whom. The results emphasize the importance of designating spaces with and for Indigenous patients, and they highlight the wholistic benefits of land-based treatment for both clients and staff alike. Transforming hospital spaces by implementing and valuing Indigenous knowledge sparks curiosity, increases education, affirms the efficacy of traditional healing treatments as a standard of care, and enhances the capacity of leaders to support reconciliation efforts.

## 1. Introduction

It is through Indigenous knowledge systems tied to the land that Indigenous Peoples have survived and thrived for thousands of years [1]; however, forced disconnection through policies rooted in colonization, racism and oppression has disrupted and weakened connections to the land, water, and natural world [2]. Processes of environmental dispossession have had profoundly negative consequences for the physical and mental health and well-being of Indigenous Peoples, as they reduce or eliminate access to the resources, knowledge and connection that underlie Indigenous knowledge systems [2,3,4,5]. As Indigenous Peoples and communities seek to recover from colonial violence and the impacts this has had on health and well-being, there is a corresponding resurgence of healthcare practices globally that are rooted in traditional knowledge systems tied to the land and connected to language, known practically as Traditional Healing (TH) [6,7]. This is largely in response to the knowledge that cultural wounds demand cultural medicines ([8] p. 147). TH practices are the original source of Indigenous healthcare [9] and involve the use of herbal remedies, healing ceremonies (often combined with specific land-based ceremonies such as the Sweatlodge), and other rituals to promote spiritual, mental, physical, and psychological well-being [9,10,11,12]. To accommodate the inclusion of TH in the provision of healthcare for First Nations, Inuit, and Métis patients, stark, sterile, and clinical spaces reflective of mainstream healthcare environments need to be transformed.

Within Canada, there is a growing Indigenous-led (Indigenous Peoples in Canada refer to First Nations, Inuit and Métis people, who are the original people and caretakers of Turtle Island, long before Canada was colonized by the British and the French) healthcare movement derived from a wholistic view that understands that healthcare combines the physical, mental, emotional, and spiritual aspects of the human being, seeing them as interrelated and interconnected [11,12,13]. A wholistic view is described as “introspective, inward, subjective, and relational in its seeking to live with and to be in harmony and balance. Indigenous wholistic wellness and healing methodologies intentionally engage meaning and relationality in teaching, learning, and healing processes; engaging the spiritual, emotional, mental, and physical as a way of coming to see, to relate, to know and activate that relationality in the living of life” [13] p. 253. Thus, Indigenous-led healthcare solutions go beyond the current, Western, bio-medical health perspective that can be far more reductionist, positivist and limited to physical and mental symptoms and processes. This movement is timely, as Indigenous Peoples’ quality of life, as measured through health disparity in every conceivable indicator of health and well-being, continues to languish behind the general Canadian population [14,15]. This growing inequity is reflected in healthcare, with Indigenous Peoples experiencing higher rates of unmet healthcare needs than non-Indigenous Peoples while reporting negative encounters and overall barriers in accessing care through the dominant system [11,14,15,16]. More options are needed to address this growing health equity gap; specifically, Indigenous populations carry a disproportionate burden of mental and physical illness, yet there is a limited decolonial body of knowledge about how Indigenous health practices can be combined into mainstream healthcare, such as publicly funded institutions and hospitals, for the purposes of healing and wellness [17,18,19].

This paper describes the creation and use of traditional healing spaces (THSs) at the Centre for Addiction and Mental Health (CAMH), Canada’s oldest and largest mental health hospital, located in Toronto, Ontario, Canada. In doing so, we draw from 22 interviews with Indigenous and non-Indigenous staff, such as hospital clinicians, physicians, and executive leaders. Building on the importance of the Truth and Reconciliation Report (2015) Call to Action #22, this research responds to an evolving healthcare landscape as hospitals transform spaces to accommodate and “recognize the value of Aboriginal healing practices and use them in the treatment of Aboriginal patients in collaboration with Aboriginal healers and Elders where requested by Aboriginal patients” [20] p. 3. Nearly a decade after this important call to action, there is a void of evidence about how THSs have been taken up in Canadian hospitals, what they look like, and if they have been effective in supporting healing among Indigenous Peoples. Through examination of three unique THSs within CAMH, this study seeks to understand how these spaces came to be, what they look like, who uses them and for what purpose.

Drawing from the discipline of human geography, and specifically the geographies of Indigenous health, the theoretical organization of this paper sits at the intersection between the natural environment (the land), Indigenized spaces of care (THSs), and healthcare environments (hospital). Thus, the format will describe how and why THSs can be used to support the delivery of TH within CAMH. In the subsequent sections of the introduction to this paper, we begin with an explanation of Indigenous healing practices and the subsequent dispossession of these practices in Canada, located within the context of contemporary Indigenous mental health. As a research study rooted theoretically in the geographies of Indigenous health, the subjects focus on the inter-relationships between the health, well-being, and places occupied by Indigenous Peoples and communities.

### 1.1. Indigenous TH and Connections to Place

It is through a longstanding connection to place that identity is formed and a deep ecological knowledge about living in balance is derived wherein “learning from the ‘Land’ interconnects with learning from the natural world, which survival is dependent upon” [2] p. 155. Across Turtle Island (North America), and documented specifically within Canada, processes of settler colonization have criminalized and sought to eradicate TH practices, replacing them with colonial structures of institutionalized and sterile hospital-based care, resulting in the establishment of a mainstream healthcare system far-removed from the land [16,19,21]. These ongoing processes have interrupted and replaced a knowledge base amassed over millennia, from knowledge embedded in Indigenous stories and histories, cultural practices, and languages, and reflecting a spiritual relationship with and respect for the natural world [4,18,22]. Therefore, the concept of land will hereafter reflect complex Indigenous interconnections that are spiritual, relational, and multi-dimensional in nature, going far beyond a conception of land as a barren physical space [3,4]. In 2020, Allen et al. emphasized that “perhaps most importantly, traditional knowledge and Indigenous spirituality hinges on the maintenance and renewal of relationships to the land. Indigenous land bases and the environment as a whole remain vitally important to the practice of traditional healing” [11] p. 180. Detailed within the national framework for Aboriginal and Torres Strait Islander health, the inter-relationships to the land are identified as central to social and emotional well-being, wherein disruption of this connection results in ill health [23] p. xxiv. TH is healthcare, where “transformative healing processes depend on the invocation of a shared cultural schema or symbolic system. This often involves extensive communication between patient and healer and, commonly, some form of education or instruction. A healer is typically engaged to spark the process and to periodically maintain it through ceremony or other treatment procedure” [13] p. 252.

#### 1.1.1. Environmental Dispossession and the Replacement of Traditional Healing Systems by Hospitals

A growing body of Indigenous research connects processes of colonialism and land dispossession to profound health and cultural impacts, arguing that cultural wounds require cultural medicine [1,3,4,24]. Therefore, we recognize the critical importance of introducing Indigenous healing practices and spaces into biomedical environments [8]. TH practices were first interrupted in Canada through policy, being criminalized within the Indian Act of 1876. Without access to the healing modalities supported through TH and ceremonial practices that were banned under the Indian Act until 1951, mental illness and the negative impacts of trauma proliferated amongst Indigenous populations [9]. Given the severity of the punishment and the residual fear associated with openly practicing ceremonies, compounded by the loss of traditional knowledge at the community level, access to TH remained limited for many decades following the 1951 legislative amendments. The resurgence of traditional knowledge systems and the implementation of TH practice in public spaces, especially colonial institutions, is a relatively new and growing practice over the last 30 years [6,19,25]. Since the time of the Indian Residential School System, institutional spaces have been widely characterized as a tool for assimilating and civilizing the original people of Turtle Island, manifesting in a harmful era where Indigenous Peoples were confined within these closed environments, far removed from places on the land [15,16,18,19]. Despite this negative history, institutional spaces continue to be used widely as purveyors of healthcare, education, and justice [19]. Specifically, hospitals have been called out as unsafe spaces for Indigenous Peoples [5,15,16]. Yet, within Canadian healthcare systems, hospitals are a foundational space for the delivery of healthcare services, and they are built predominantly on a Western, bio-medical system that excludes Indigenous TH and access to the land. Thus, important determinants of Indigenous health, such as cultural continuity and connections with the natural environment, are largely absent or ignored in these places, limiting access to the wholistic healthcare practices needed to improve Indigenous Peoples’ health outcomes.

#### 1.1.2. TH and Mental Healthcare

The bio-medical healthcare system, and subsequent hospital sites, have neglected the needs of Indigenous Peoples’ mental health [21,22,26]. Despite advances in modern healthcare, including increased social services, dedicated resources for research and the implementation of abundant Indigenous health programs, very little improvement in the overall health and well-being of Indigenous Peoples has been documented. The limits of improvement point to ongoing structural and systemic disparity wherein the current systems are not adequately equipped to attend to these inequalities [1,8,9,27]. Disconnections to culture through dispossession have been linked with a diminished cultural identity and poor mental health and well-being [21,23,28,29]. Outcomes include a sickness that begins within the spirit: spiritual and cultural imbalances manifest through interconnections in health, permeating sickness through to the mind, emotions, and body [12,23,29,30]. This phenomenon is often described as a persistent and pervasive form of intergenerational trauma, whose impact can be described as “cultural wounds” [8,25,30]. True healing includes re-establishing a balance between the four realms of the physical, mental, emotional and spiritual [30] wherein “the provision of mental health and addiction treatment and services to Indigenous patients must be fundamentally restructured to include Indigenous values and realities” [6] p. 14. The cosmology of Indigenous Nations is based on balance. All the ceremonies are a re-enactment of creation and a seeking of the restoration of balance in the individual, family, community, and natural world.

#### 1.1.3. Transforming Healthcare Spaces

Hospitals such as CAMH are systematically engaged in updating and transforming their structures to reflect a more contemporary understanding of healthcare delivery [31]; therefore, it is timely to consider the space needs related to Indigenous Peoples and Indigenous healthcare. We are also in a time of tension and change, as Canadians struggle to enact the Calls to Action from the Truth and Reconciliation report [32]. Healthcare has been a focus of targeted Calls to Action because there has been so much racism and harm proliferated through these systems, and there is strong evidence that suggests that places of care, such as hospital institutions and mental health institutions, continue to lead the way as environments where Indigenous Peoples experience harmful interactions [33,34]. In the Canadian context, colonialism or colonization is a process of erasing Indigenous Nations and replacing them with a new, dominant society. The process of decolonization has been defined by Corntassel as “embracing a daily existence conditioned by place-based cultural practices” [35] p. 89 and “involves unlearning… and embracing messiness and tensions in the process of transformation” [36] p. 17. Through this movement of the decolonization of healthcare, community-based Indigenous health spaces are leading the way by combining both traditional healing systems with Western, bio-medical approaches to care, building the evidence base needed to see these practices extended to mainstream healthcare places such as hospitals.

#### 1.1.4. THSs in a Mental Health Hospital Context

To offer TH practices effectively, institutional spaces must be transformed to facilitate access to or connections with the natural environment, which may include the burning of plant medicines, often referred to as smudging [19,28,33,34]. In 2022, Asamoah et al. presented their systematic review of Indigenous Traditional Healing programs within the mainstream healthcare systems in Canada, Australia, and New Zealand, concluding that there are few examples [6]. In fact, there was only one mental health hospital-specific example drawn. The only mainstream mental health hospital that offers traditional healing as the main form of treatment applies the Whanaungatanga model of care, as adopted by the Tauranga Hospital located in the Western Bay of Plenty in New Zealand [37]. This example documents the benefits of “culturally safe care” and the efficacy of traditional healing within the mental health context [36]; however, it does not distinctly speak to the spatial aspects of spaces needed to combine these practices with Western medicine. In Canada, a hospital in Sioux Lookout, Ontario, is a remarkable example of a healthcare facility designed from the ground up to offer a traditional medicine program to the predominately First Nations patient population. This institution has been documented as an important site of access to Indigenous Traditional Healing services, describing positive health impacts and culturally responsive healthcare [23,37,38], yet with a very limited focus on mental health specifically and limited perspectives on the physical spaces for these practices.

## 2. Case Study and Methods

CAMH is Canada’s largest mental health and addictions teaching hospital and a world-leading research centre in this field [39]. CAMH combines acute and primary care, research, education, policy development and health promotion to help transform the lives of people affected by mental illness and addiction. CAMH is fully affiliated with the University of Toronto and is a Pan American Health Organization/World Health Organization Collaborating Centre. Since 2000, efforts to improve mental health care for Indigenous patients have been led primarily through Indigenous staff of the Aboriginal Service (now known as “Shkaabe Makwa”), “to create a culturally responsive service to meet the mental health and addictions needs of First Nations, Inuit and Métis clients at CAMH” [31]. As recognized in the land acknowledgement located in every building of the hospital, the “CAMH has a very special relationship with the land at its main campus at Queen Street West. In historic records, the land has been described as the Council grounds of the Mississaugas of the Credit. This preferred location allowed for camping for the hundreds of Mississaugas who travelled by canoe to conduct the governance of their Nation, trade, and engage in negotiations for treaty making and land transactions” [40]. CAMH was selected as the site of this in-depth case study as it is leading and innovating mental wellness care through expanding the Aboriginal Service and transforming clinical spaces within the hospital to offer Indigenous cultural care and TH, known as THSs. In summary, CAMH was selected for two distinct reasons: (1) both the history and contemporary relationships between the primary hospital site and the relationship between Indigenous populations are unique and rooted in place as demonstrated through the historical map in Figure 1a; and (2) the research team has had a relationship with this hospital over time and as such is positioned to respond to the research needs identified by the Indigenous Research Circle (IRC).

CAMH has three distinct THSs that are examined within this case study, pictured on the map in Figure 2: the Ceremony Grounds (CG), housing the Sacred Fire, medicine gardens and the first operational Sweatlodge for patient treatment at an Ontario hospital [31]; a Ceremony Room (CR), and a virtual care space, known as the ECHO Ontario First Nations, Inuit, and Métis Wellness Room (referenced as the “ECHO RM” hereafter). It is these three unique cultural spaces, considered together as the “Traditional Healing Spaces”, that are the primary focus of this case study.

### 2.1. Methods

This study employs an inductive, relational and decolonial approach to research [41], including qualitative inquiry methods guided by a dedicated Indigenous Research Circle (IRC) and led by an Indigenous research team from Western University. This research is part of the Ph.D. project of the first author, who is of mixed ancestry (Cree and Métis) and who has worked in the Indigenous health field for the past two decades. During her tenure, she collaborated extensively with health organizations, including CAMH, to advance Indigenous health equity. The last author is an Anishinabe scholar from Biigtigong Nishnaabeg and is the academic supervisor to the first author. The IRC was central to developing the process for this study, and collectively the group devised the names to be used in referring to each of the three THSs. Data were collected through 22 in-depth and semi-structured interviews with current staff, who include physicians, clinicians, and executive leadership at CAMH, from the care provider perspective. The semi-structured, virtual interviews were co-designed and led by an Indigenous researcher (over Zoom video conferencing 5.0 software) with current Indigenous and non-Indigenous staff at CAMH.

Drawing on the geographies of Indigenous health, this case study investigates a colonial institutional “place”, describing and characterizing the configurations and uses of TH at CAMH. This information serves as a useful way to contextualize the processes of decolonization and reconciliation within the broader hospital environment, building on a perspective from the care provider. The term “case study” refers to both a method of analysis and a specific research design for examining a problem, both of which are used in most circumstances to generalize across populations [42]. Within a case study approach, the subject of analysis is bound by time and place as it seeks to explore peoples’ experiences of and with one of three THSs within CAMH, which have only existed for 8 years, except for the Ceremony Room, which was officially opened in 2008. The data, presented as an in-depth case study, enable one to extrapolate key themes and results that help predict the future transformation of space, illuminate the practice of privileging space for TH, and/or provide a means of understanding the research problem with greater clarity. Knowledge gained from communities is both local and specific to a given research effort, but it is also global in terms of its history and potential impact [43], as case example research can also be designed as a comparative investigation that shows relationships between two or among more than two subjects.

The study participants were identified by the IRC based on the criteria that they must have had some direct interaction and/or experience with at least one of the three identified THSs in the scope of this study. Table 1 indicates the characteristics of the interviewees. The IRC is composed of a group of primarily Indigenous leaders (plus one non-Indigenous leader) working at CAMH (*n* = 5), who convened this circle for the purposes of guiding this study from the beginning through to the end.

Members of the IRC were also included as research participants as they were the most familiar with the evolution of Indigenous care at CAMH. The IRC served as a key support to the research team, who were outsiders to the hospital, in establishing relationships within the hospital, completing ethics, and providing ongoing support and advice about the research goals and process. These elements are central to conducting responsible research with Indigenous Peoples and communities [4,44,45]. The IRC was central to implementing a decolonial methodology that was “community-based” [41,43], as this Circle was comprised of Indigenous staff supporting Indigenous priorities within the hospital and who themselves are also members of the larger Indigenous community in Toronto. It is through this relational accountability [46] that decolonized research principles were established and then enacted through a formal relational research agreement between the research team from Western University (and the co-authors of this paper) and the IRC, predating institutional ethics. This process ensured that the study design was informed by the IRC and that the research would be relevant and responsive to the needs of the Indigenous patients and critical for informing the practical and policy level work that CAMH itself performs. This was founded on the principle of reciprocity, where mutual benefit for both the lead researcher (contributed to her Ph.D.) and the IRC was a key consideration that guided decision-making and research direction. At all costs, this work was developed in such a way that the results will enable CAMH to engage in systems transformation for better patient outcomes. These connections and relationships are central to the design and subsequent theoretical framing of the research process and guide the thinking behind the doing, enabling the researchers to view themselves as part of the research, recognizing the inherent value of “privileg[ing] Indigenous knowledge systems as the source of epistemological and methodological relationships in transforming what and how we can know about the world as a foundation for deriving new [insights and] knowledges about the world that may challenge a Western, Eurocentric understanding of the same research question” [41] p. 11.

### 2.2. Analysis

Analysis of the 22 interviews involved a combination of thematic and descriptive analysis. NVivo 12 qualitative data software was used to organize the interview transcripts, enable coding, and support an inductive thematic analysis. The researchers undertook a constant comparative method to organize and identify the main themes and narratives. This process was performed by the lead author only and was part of her doctoral research study. Notably, the IRC was also involved in the data analysis. In particular, the lead author and last author worked with the IRC to review and place the thematic analysis within the social, political, geographical, and historical context of the hospital and the wider urban Indigenous community. Since both Indigenous and non-Indigenous perspectives contribute to this dataset, another method used was to compare and contrast the themes, seeking to identify any parallels and/or differences coming from the many different knowledge positions that the hospital staff and health professionals bring to these interviews, which were examined as part of the analysis.

## 3. Results

There was a tremendous knowledge base amassed through the detailed interviews conducted with Indigenous and non-Indigenous staff (*n* = 22). One of the preliminary results is based on Table 1, which documented the characteristics of who was interviewed. The majority (64%) of participants self-identified as Indigenous (36% did not self-identify as an Indigenous person), and most people interviewed identified as female (77%). Of those interviewed, the staff were clustered within an Indigenous program/team (60%), with the majority serving in a leadership-level capacity (60% were either managers or directors). An additional minority of those interviewed also had a clinical designation (23%), so some of the interviewees carried more than one designation. Given the high proportion of Indigenous staff, a source of information was drawn through collective knowledge about the history of how the THSs came to be. A prime example of how the history has been documented can be recognized through Table 2, which is populated by data coming from the participants and details the date each THS was established, and it also details the original use of these spaces, as there is no single report or document that carries this level of details and specificity.

The data also helped to capture a chronological timeline that documents the evolution of the Aboriginal Service, including the establishment of the THSs, and key Indigenous-led programs and reports throughout the hospital’s history (Figure 3).

Additionally, when asked about the history of Aboriginal Services, one staff interviewee reported an important context that has not been formally documented and related to how it was established at CAMH in 2000. At that time, traditional teachings and cultural knowledge were shared with clients at CAMH by Elder-in-Residence Asin Vern Harper baa (1936–2018) through TH practice such as the smudging ceremony, singing, and drumming. Prior to the creation of a dedicated THS on campus at CAMH to conduct Sweatlodge ceremonies, Vern Harper baa (see Figure 4) would bring clients to his own personal THS, located about one hour east of Toronto. Regrettably, this practice was short-lived because “as he aged, he became quite frail, and they didn’t make as many trips to bring clients out there… [thus] there were discussions always about doing sweats at CAMH” [Staff 1: p. 4].

### 3.1. What Do the Traditional Healing Spaces (THSs) Look like and Who Uses Them?

The THSs at CAMH grew from the recognition that Indigenous-specific mental healthcare service delivery requires a connection to a land base [9,10,12,13,20]. The growth of the THSs was linked to the growth of the Aboriginal Services, with the launch of many more Indigenous-specific roles at CAMH. “Early attempts to create a THS and secure funding to build a Sweatlodge at CAMH were unsuccessful” [Staff 1: p. 4]. In 2008, the CR was established (see Figure 5) to meet the need to incorporate ceremony and culture into patient treatment, demarking the first of the three THSs examined in this study.

As of 2023, the three THSs at CAMH are identified using the following names and abbreviations: the Ceremony Room (CR), the Ceremony Grounds (CG), and the First Nations, Inuit, and Métis ECHO room (ECHO RM). When the interviews were conducted, the THSs were referred to by the names indicated by the IRC and labelled throughout the data collection process (including the interview template, letter of information, questions, etc.) as the CR, CG and ECHO RM; however, an interesting finding surfaced during the interviews as there was no consistent reference to these THSs. There were, in fact, 32 unique names used to refer to the different spaces (see Table 3). When reviewing the types of names used to identify each of the spaces, there was a diversity of references used (e.g., the CR was called “room 116”, or “60” for 60 White Squirrel Way, the address of the building the room is in), or a reference to what happens within the space (e.g., the CR was referred to as “the smudge room”, referencing the fact that this room has a separate ventilation system to enable the burning of plant-based medicine like sage and sweetgrass). There was no indication that there was a specific name, brand or common identifier used consistently by all the participants.

Table 4 indicates that the most accessed THS by the study participants is the CR, followed closely behind by the CG. In addition, 36% (*n* = 8) accessed all three THSs. Many of the remaining participants accessed at least two spaces: 45% (*n* = 10) accessed the CG and the CR, while 9% (*n* = 2) accessed the CG and the ECHO RM. The remaining two participants (9%) accessed only one THS: the CG (virtually) and the ECHO RM. As one participant involved in all three spaces indicated, “I’ve always been progressive in being a part of the digital world and virtual care, all these things I put my hands on led me into each of these spaces in some way shape or form” [Staff 1: p. 9].

#### 3.1.1. Description of the Ceremony Room (CR), est. 2008

Based on qualitative descriptions drawn from the interviewees, a narrative about the history and use of the space emerged, and we have ordered its evolution over time. What started out as a stark and typical hospital clinical room, with long tables and chairs, has “evolved slowly over the years… [and grew from] the need to offer clients a Sacred Space that they felt was theirs, and somewhere where we could do traditional ceremonies and healing” [Staff 1: p. 2]. Thus, the Indigenization efforts concerning this space were propelled by the clients themselves, who first removed the tables, placing the chairs in a circle, and “the Indigenous artists engaged by CAMH painted a beautiful design of the four sacred colours on the floor, and that started to show that it became an Indigenous space: by putting that Indigenous artwork, painted on the ground, it became permanent” [Staff 1: p. 3]. It should be noted that the original piece painted by patients on the floor is not the same mural that exists today: to make it permanent, Indigenous artists were contracted to paint the medicine wheel in 2015. Staff also offered significant physical descriptions of the spaces throughout the interviews, helping to detail the descriptions contained herein.

The CR is a small, rectangular room, measuring 4 m by 5 m (13 by 17 feet), with a row of windows along the north and east walls, with one doorway located on the south wall. The walls of the room are paneled vertically in cedar. Indigenous artwork hangs on the walls. There is a row of locked built-in cabinets on the north wall, housing supplies and traditional medicines. There is a tile floor, with a hand-painted mural in the centre of the room, designed and painted by Indigenous artists in 2013. The mural depicts a Medicine Wheel, featuring the four directions using black, white, yellow, and red, delineated by circles/stones, and with blue threads of water weaving through the circle. There is no table but rather stacks of chairs that can be arranged in free form, usually in a circle configuration around the mural. There is a projector on the ceiling, and a retractable screen that can be used to project images/films. The hidden but defining feature of this room is the separate ventilation system that cleans the air and enables the burning of plant-based medicines, known hereafter as “smudging” [26,47].

Interviewees noted that the CR was built in 2008 as a gathering space for Indigenous clients of the Aboriginal Service, which at that time hosted a 21-day Indigenous men’s addictions treatment program. In 2009, external funding was secured by the ABS through the Aboriginal Health Transition Fund (Province of Ontario) to retrofit the room with a separate ventilation system, which “had a vented hood fan built into the ceiling system to be able to allow for ceremonial practice of smudging and sacred pipe, to burn sacred pipe” [Staff 1: p. 2–3]. The space was developed for distinct cultural practices, including the ability to sit in talking circles, engage the Spirit in healing ceremonies, drum, sing and receive traditional teachings. The CR has been described as a more calming, less clinical therapeutic environment for group sessions, where “just seeing some of the actual items and artifacts, and just the way the space is set up is quite different, obviously, from some of [CAMH’s] other clinical spaces… there’s the ability to Smudge and so I think there has been a relaxing of certain rules or restrictions in the hospital to allow for some of these traditional practices” [Staff 2: p. 2]. People gather in the CR room to participate in traditional healing ceremonies for several different reasons—from client group sessions to staff meetings.

With the demands on accessing the CR growing, clients’ participation had to be monitored and limited as “it’s a small space, it’s not a large space. At one time, my circles grew to over 20 different people, and it became unmanageable in there…it wasn’t large enough to accommodate all those different types of chairs, so there was no choice, I had to start limiting clients to 10” [Staff 1: p. 4]. In 2010, as a response to the demand on the space, the CR was designated, under hospital policy, as an Indigenous space that could only be booked and accessed through the Aboriginal Service. This policy was created to protect and prioritize the space for Indigenous clients. Even with the shift to decentralize room bookings, the use of the CR still extended beyond Indigenous client use and it was frequently used as a meeting space for Indigenous/allied staff and as an educational space for key visitors to CAMH. Currently, the CR is accessed and used by six key groups: Indigenous patients; Indigenous staff; the ABS team, the Shkaabe Makwa Team; CAMH Reconciliation Working Group, and members of the Indigenous Caucus formerly the Aboriginal Caucus (First Nations, Inuit and Métis staff from various programs and services across CAMH, who meet five times a year and actively support the National Day for Truth and Reconciliation during the month of September and National Indigenous History Month in June each year by creating learning opportunities for staff members and the public [48]).

#### 3.1.2. Description of the First Nations, Inuit, and Métis (FNIM) ECHO Room (ECHO RM), est. 2015

In 2015, the First Nations, Inuit and Métis ECHO room was transformed from a classic clinical space into a dynamic and customized THS for the delivery of virtual healthcare and equipped with a portable air purifier to absorb any smoke from smudging. To facilitate ceremonies during sessions, “the Elder is always asked to open up with a prayer and smudge before every ECHO session” [Staff 1: p. 6], and the portable smoke machine (air purifier) must be used and turned on while Smudge is being burned. Established in tandem with the launch of the First Nations, Inuit, and Métis Ontario Wellness ECHO program (ECHO Ontario Mental Health is a virtual training and capacity-building model that supports healthcare providers in delivering high-quality, evidence-based mental health and addictions care in their local communities), unique funding was procured through donor funds by the Director of ECHO to Indigenize the space. Members of the Indigenous staff programs and caucus were engaged in the decision-making around the elements included in the space. The features of the ECHO RM, pictured in Figure 6a centre around “a beautiful table made out of a gigantic tree, it’s a cut slab, I think it could be Maple or Oak; it’s a beautiful slab with a decorative art piece in the middle with some glass and some rocks in it” [Staff 1: p. 6]. Visible to participants online, who join in via virtual telehealth screens on the wall as pictured in Figure 6b, include Indigenous artwork shrouded by a custom, blue-painted wall colour (Nicole Tilly blue) chosen to solicit a calming effect of water and adorned with the First Nations, Inuit, and Métis ECHO logo. “These are beautiful things to have on the lens for the communities of Ontario, the Indigenous communities, to see that we’re making an effort to Indigenize our space” [Staff 1: p. 6]. 

The ECHO RM is used for two primary reasons: (1) as a formal meeting space for the First Nations, Inuit, and Métis ECHO program to hold clinical group meetings with clients in remote locations (i.e., virtual mental health services in northern Ontario); and (2) to support Indigenous approaches to connection and treatment planning, offering a space for staff and clients to share personal and lived experiences. Beyond client support, the room is also sought after for additional uses, including the ongoing education and training of staff, particularly those who support First Nations, Inuit, and Métis virtual health consultations in remote regions. More recently, the space has also been used to connect with other Indigenous ECHO teams across the province and to meet with the government program funders. “I think [this space is] very meaningful… it sets the right tone for whatever is happening in those spaces… it shows that effort has been put in before and that people really care to make this a success” [Staff 3: p. 5].

As Table 3 indicates, only 50% of study participants had accessed this space, and it is unclear whether the ECHO RM is viewed in the same capacity as a THS, wherein the CR and CG are often grouped together as a Traditional Healing Space. One participant noted that the ECHO RM was often used by other non-Indigenous clinicians (i.e., other provincial ECHO programs) and that sometimes the room was not available for Indigenous clients when needed. In addition, several participants noticed tensions around sharing the space with other programs/staff, as there was an additional burden placed on Indigenous staff who come to use the room and find items misplaced or who must clean up after other users of the space. The room is integrated into the hospital’s main booking system, meaning any staff can request and book the room. There is an existing hospital policy regarding the use of the ECHO RM through the Smudging Policy to enable the burning of sacred medicines, stating that to do so requires the use of the portable air purifier.

The limits of the space have been documented, including the small size of the room (measuring 11 m^2^ or 120 ft^2^) and the lack of natural light (intentional in the original design to support the use of virtual screens) as it has no windows and is in the basement of the 33 Ursula Franklin Way site. In addition, the space contains no separate climate control, so could be “very warm, sometimes uncomfortably so… [and] using the smoke machine can be quite noisy” [Staff 4: p. 3], potentially creating difficulty for those who may have challenges hearing.

#### 3.1.3. Description of the Ceremony Grounds, est. 2016

“The Ceremonial Grounds, it’s not even in the ‘heart of the city’, but it’s the heart of the city. It’s such a beautiful space… the city just falls away… it’s as good as you can get to being out of the city, within the city” [Staff 4: p. 4]. In 2015, a tract of land (measuring 6.7 m wide by 38 m long), located behind the building at 10 White Squirrel Way, was allocated to the development of a Sweatlodge by CAMH Redevelopment Department. This space became designated for the third THS at CAMH, known as the Ceremony Grounds (CG), which house the medicine gardens, Sacred Fire and Sweatlodge (Figure 7). The land has been “spiritually consecrated”, meaning that it has been confirmed through ceremonies that the Spirit has given permission to build on this land [49]. Subsequently, two ceremonies took place to enact traditional protocols to use the land for healing purposes, including one for “Mother Earth, seeking permission to build and for Her blessing on the healing that lay ahead for clients… [and] the second ceremony was for the Ancestors—to enlist their guidance and to ensure that they were at a state of rest given the long history of CAMH on this land, beginning as the Provincial Lunatic Asylum in the 1850s” [49].

CAMH was able to support TH for years prior to this launch, with the 2000 hiring of a staff Elder (Figure 4) and the 2012 hiring of a full-time Cultural Resource Worker and full-time Traditional Healer, which has been characterized as the formal launch of “traditional healing doctoring” [50] as treatment for Indigenous patients at CAMH. Thus, prior to the CR and CG being established, TH practices were not new to CAMH. However, based on documentation (Figure 3), TH practices as treatment were amplified through dedicated strategic planning led by Indigenous staff hired specifically to advance the Indigenization and decolonization efforts detailed in the 2016 report Guiding Directions [51]. Access to the Traditional Healing Spaces requires strong, trusting therapeutic relationships to be formed; additionally, “if you are a patient at CAMH …you still need to do a lot of self-advocacy to your Clinician, because if your clinician doesn’t know enough about the access at CAMH in Aboriginal Services… people are missing out on treatment—cultural treatment” [Staff 5: p. 3].

The features of the CG are all co-located within a narrow tract of grassed land located between the wall of one of the original buildings and the original wall of CAMH (200-year-old stone wall), located behind 10 White Squirrel Way (see Figure 1). This piece of green, grassed land is narrow and elongated—approximately the width of a single car garage and over one-third of a football field long. There are two separate fire pits: the Sacred Fire and the Sweatlodge Fire, dug into the ground and surrounded by stones and incorporated into treatment, ceremonies, meetings, and education [49]. Located along the wall of the hospital is a shed with tools used for gardening, groundskeeping and building the Sweatlodge. The sides of the space are lined with cedar trees, an important source of medicine for the Sweatlodge ceremony. The Sweatlodge is located centrally and is constructed of willow tree, harvested outside of Toronto and brought in specifically for the ceremony along with the large stones to be used in the lodge once heated in the Sacred Fire. At the north end, there is a large, locked, gated entrance restricting access to the space. There is no access to bathrooms from the Grounds, meaning that clients need to exit and re-entre buildings along White Squirrel Way. The west side of the land includes the remnants of the original asylum’s stone wall (Figure 8b) and many old-growth trees. On this side, there are two condominiums that overlook the space; however, it is still private, as people on balcony would have to lean over to see activities taking place on the CG. The space is used by many different groups as “the layout is incredible because it lends to a fairly large space; it’s rectangular in shape, which allows for many people to gather and form many, multiple private activities to be concurrent” [Staff 4: p. 6], and it can be used as overflow space for the CR once client numbers for group sessions grew beyond 10 people. Additionally, people gather to engage in multiple activities within the space, including use of the two Sacred Fire pits and the ability to have burning fires. The CG is designated, under hospital policy, as an Indigenous space, and it can only be accessed and booked through the Aboriginal Service.

### 3.2. Key Findings Related to the THSs

Though two of the spaces are reserved and designated for Indigenous patients of CAMH and can be booked and accessed only through the Aboriginal Service, the CG and CR have been widely used by staff of the ABS, Shkaabe Makwa and members of the Aboriginal Caucus. These findings reinforce the importance of relationships with Indigenous staff as key knowledge holders, who have been engaged in many ways since inception, including participation in ceremonies. For example, upon opening the CG, Indigenous staff served an important role in hosting the ceremony, given that building a Sweatlodge “requires a lot of specialized knowledge, and support” [Staff 1: p. 7]. Thus, it is not uncommon for Indigenous staff to be called upon to “engage with Aboriginal Services team and help support with National Indigenous Month, which means, maybe we’re supporting the Elder with setting up the Sweatlodge, or maybe we’re joining and having potlucks and doing all sorts of different stuff across the hospital” [Staff 5: p. 3]. As documented by a study participant, “Having ceremonial spaces within CAMH really affirms our connection to the land base and therefore who we are as Indigenous Peoples” [Staff 6: p. 5]. Finally, one unique aspect of the CG is that there is a broader policy to enable use beyond CAMH’s patients/staff, to include broader Indigenous community/families and guests visiting the hospital for educational purposes.

#### Thematic Findings

One theme that surfaced concerns access; specifically, how an Indigenous patient at CAMH gains access to TH or a THS. The sub-themes that surfaced included the variability of client access, noting that “clinicians still carry a lot of privilege and that they’re all gatekeepers, and so unless the right forms are complete, patients aren’t able to access the space in the way they need it if those aren’t completed” [Staff 5: p. 3]. Within this theme of access is “privilege”. For example, privilege to access Traditional Healing/Medicines has been noted by one participant who recalls: “If I want to go and participate in Traditional Healing, I can go and do that on the land—I have access to that. If I need it, I can go and I can get medicines, but then I think what I struggle with is that ‘this is a privilege’, if I think about the patients” [Staff 5: p. 3]. While the CG space was created centrally, “making it possible to address healing of all four areas of healing—the physical, mental, emotional and spiritual all at once” [Staff 6: p. 5], access was limited to times during working hours when staff could facilitate this process, which is very labour- and resource-intensive. The impact on participants has been described by one staff member in their role in supporting the Sweatlodge: “I’ve seen this have a remarkable impact on people, just in my role as Fire Keeper; when you’re silent, you watch and you can see how people interact before and after ceremony…it’s more relaxed, less isolated, you know, more of a sense of community among them, even though some of them may have met … a few hours prior” [Staff 7: p. 4].

Great care is taken to prepare the CG for the Sacred Fire and Sweatlodge ceremonies, as paraphrased from the “Sweatlodge Teachings” with Traditional Healer Kawennanoron Cynthia White (2021): “The Sweatlodge (a purification lodge) is constructed from a willow frame covered with a tarp (a black canvas tarp). Participants of the Lodge are asked to help prepare the Lodge before entering, this includes harvesting and laying out Indigenous sacred medicines (cedar boughs) in a circle around the lodge to create a ‘sacred space’. The Lodge space, Sacred Fire, and elements (large stones that are referred to as “grandfathers” and/or “grandmothers”), and buckets of water that will be used in the Sweatlodge are smudged by burning Sage (a sacred medicinal plant) and the smoke is spread over the space and items to purify them and to create this sacred space” [50].

Indigenous structures like the Sweatlodge are critical for supporting connections to identity, safety and belonging, and this requires the use of the Sacred Fire, natural elements like rocks, and access to a land base [25,30]. “These kinds of spaces really help remind us about questions into the broader aspects of peoples’ lives, beyond their treatment…so, not just the service they’re receiving and getting beyond transactions-of-service and really bringing people into places and spaces that honour their whole, human being” [Staff 2: p. 5]. In addition to creating more inclusive and connected space for the patients, the THSs at CAMH were reported to also be meaningful for staff at CAMH, with Indigenous staff citing that they relied on the spaces as a place to recover, reflect and decompress from their own experiences of trauma within the hospital. “Those are spaces are designed for our people to feel less institutionalized. It, whether they identify as Indigenous or not, the space speaks to one’s heart and soul in ways that other things can’t even come close to… many individuals who reside in the city of Toronto, who are challenged in their wellness, face being disconnected from the land, and the Ceremony Grounds offer a small connection to the land” [Staff 4: p. 6].

## 4. Discussion

The key findings of this study demonstrate three specific areas of interest, access vs. privilege, designation vs. protection and naming vs. claiming. The findings from this study show that the THSs have grown far beyond their original purpose and intention. At CAMH, the current use of the THSs has extended beyond enhanced Indigenous patient treatment to include regular use by Indigenous staff to support their own mental wellness and healing. The data allude to the structural violence and racism experienced by Indigenous staff at CAMH, who may encounter resistance in supporting Indigenous TH work within a colonial institution that privileges a Western, bio-medical model of care, and they require access to culturally safe spaces and TH to recover.

As a case study, this research provides preliminary analysis on the phenomena of THSs within a mainstream healthcare environment, seeking to build knowledge in an area of scholarship that is still emerging. The findings indicate that while staff access THSs to recover from colonial violence and trauma, there are other important uses of space for staff, including opportunities to access ethical spaces of engagements that are important for advancing Indigenous cultural safety and humility education. Education and addressing the full history of Indigenous and non-Indigenous relations in Canada can be difficult conversations to have with settler, non-Indigenous Canadians. Research has indicated that successful anti-racist education and transformative learning requires ethical spaces [52]. While education was a regularly cited reason for using the THSs, this may overburden resources designed for patient use, suggesting the need for additional spaces for decolonizing and Indigenous education. The data align with the recommendation to support ongoing critical engagement and educational processes within and across CAMH. For example, for important discussions on reconciliation, decolonization, Indigenous knowledge systems, Traditional Healing, “Indigigogy” [Staff 2: p. 2], Indigenous Cultural Safety, racism, and structural barriers, the THSs offer a safe and comfortable environment that is conducive to this level of transformative education [15,16,18,33]. However, limits in access to the spaces may be problematic, as they are predominately designed for the delivery of TH and direct patient care.

Only two out of the three THSs are designated specifically for Indigenous patient use (CR and CG). Even with this designation, observations within the data suggest that the CR is very popular and difficult to book. This is an indicator of two things: (1) that the need for the CR has increased, and (2) that there is a desire to access spaces that have been transformed to enable TH. Limits in accessing the space have been documented, including the requirement for designated staff to unlock the CR and accompany patients requesting to use this space. Additionally, there are limits in access to clients/patients who have been specifically registered through the Aboriginal Service, and/or have the explicit support of their treatment team. At the time of the interviews, the Aboriginal Service was a clinical addictions program, so programmatically, patients had to be identified as a substance user to qualify.

The CG and CR are designated, under hospital policy, for Indigenous use. However, the ECHO RM is not a “designated” Indigenous space and it is desirable for many other ECHO team members to use space for their own meetings. Distinction related to designation matters when viewing equitable access and prioritization. Indeed, in all cases, externally acquired, specialized funding was procured to develop each of the three THSs at CAMH and was not resourced internally (except for the CR cedar wall paneling and the mural fee). The original purpose of these spaces was to address limitations in the Western and bio-medical models of care that have been harming and less efficacious for Indigenous patients, and one area of concern in not protecting and designating spaces for Indigenous use is the risk that they will be co-opted for use beyond this population. When viewing access to spaces, the example of the ECHO RM is important, as it is often occupied by various other ECHO programs (non-Indigenous) and is a desirable place for staff of CAMH to meet. Staff identified that “it’s difficult to make those changes and raise awareness…it’s something that has always been a struggle, but slowly [CAMH has] been able to make and modify the rooms” [Staff 1: p. 4], which may allude to a larger issue around access. Gaining access to the actual space is another important consideration. For example, the THSs are not open spaces; they exist as locked space: “CAMH is not an open door [and one] must be a client of CAMH, or Aboriginal Services to access that medicine room and that smudging service. they always have to know who you are… a staff need to be with you at all times to access that” [Staff 1: p. 3].

The findings emphasize that the THSs have far exceeded their original intention of Indigenous client care, showcasing their potential for advancing cultural-safety education and enhancing staff and patient mental wellness alike. This latter finding indicates an important tension: the THSs were designated to improve mental healthcare for Indigenous patients and to advance reconciliation, and since they have become in-demand spaces of healing for Indigenous and non-Indigenous patients and staff, is there a risk of losing sight of this original intention? This study reveals that the THSs are unique and innovative environments within hospitals that appeal to everyone—patients and staff alike; however, recognition and designation of these spaces for Indigenous healing are critical as they serve unmet mental healthcare needs of Indigenous patients.

### Reclaiming Hospital Spaces through Environmental Repossession: Building on the Geographies of Indigenous Health and Healthcare Environments

The THS study grows from the geographies of Indigenous health and pulls from the theoretical framework of Environmental Repossession (ER), which is a foundational concept to “examine the complex and changing relationship between Indigenous Peoples’ health and the environment” [4]. To perform geographical research in meaningful ways, Indigenous geographers have embraced a relational framework stemming from the uptake of applied Indigenous community-based participatory research approaches [27,44,45,46] and evolving through the development of Indigenous relational protocols [53] that are used to advance and benefit Indigenous communities’ health and wellness [27,46]. Indigenous geographers now seek to embody Indigenous methodologies, theoretical frameworks, and Indigenist, self-determined research [4,19,44,45]. Within this promising new discipline, this study builds on a process of community-engaged research, building on an Indigenous, relational worldview and privileging Indigenous voices and perspectives from the ground, while seeking to validate relationships and experiences within the THSs. Indigenization of the space was slow, and it was identified that change happens slowly within CAMH and its systems as an institution. The exploratory research into the THSs within a hospital environment has illuminated many themes that are relevant, timely and important as more and more institutions seek to advance efforts to bolster reconciliation, Indigenization, and decolonizing efforts.

## 5. Conclusions

Our findings amplify the need for a geographical lens in Indigenous health research and seek to serve as a promising example of institutional transformation for any healthcare, education, or other institution involved in Indigenization and reconciliation efforts [19]. We can observe that the three THSs have far exceeded their intended use and it has been recommended by study participants that more spaces are required to meet the needs associated with Traditional Healing. It can be concluded that the THSs in CAMH are in fact not only meeting intended use but there are many more uses that these spaces support—for staff, patients, and other healthcare spaces. A result of this increase are the limits on what these spaces can and cannot support, where significant growth has meant that there are needs that are not being met through the current spaces.

To meet the Canadian Truth and Reconciliation Report’s (2015) Calls to Action [20], there is an ongoing and pressing need for advancing education and awareness about Traditional Healing and the value of Indigenous knowledge systems in healthcare. Creating a new space for dialogue has been called out by Ermine in 2009, who noted that these spaces create a contrast by dislocating and isolating two disparate knowledge systems and cultures [52]. Ermine emphasized that ethical spaces of inquiry are required to build common ground between worldviews, noting “there have been lots of good attempts by sincere people who have tried to build bridges, but these undercurrents are powerful and keep washing away good intentions… when we have had breaches and ruptures in the past, it is because we have failed to look at the area in between our two worlds” [52]. Tobias et al. (2013) reminded us that Indigenous “knowledge systems are still greatly being applied to help Indigenous communities and Indigenous Peoples recover from intergenerational pain…future policy development and implementation should aim to support Indigenous Peoples and communities when they decide to learn about, maintain, and build upon the knowledge amassed by their ancestors” [53] p. 3. We have also come to recognize that relationships are central to advancing this movement, wherein “Indigenous-led healing movements reflect concrete steps in the efforts to advance health equity for Indigenous Peoples in Canada” [10] p. 215. However, support for Indigenous TH practices within Western healthcare systems has been lacking, due in part to the history of colonization and differences in worldview, despite various policy recommendations that have been made [6,27,54]. Questions emerging from the data related to these tensions are built on longstanding colonial-settler tensions and include the following. Should there be a unique formula/lens for Indigenous programs/space used in CAMH? Is there a rationale for designating space for Indigenous use? How does this differ from spaces for others, and “why not unique space for all identities?” Does the visibility of space matter? Why? Additional study to attend to key questions emerging from the data can help support this emerging case to determine whether the THSs in hospitals become places of healing to counteract the impacts of intergenerational trauma instead of harm. This includes a better understanding of how we can make spaces that are appropriate, well-resourced, and meaningful for the Indigenous Peoples who use them, specifically centring the voices and stories of both Indigenous staff at the hospital and patients and enabling the system to adequately respond to the mental well-being needs of this sector of the population utilizing the services of CAMH and informed by data as a priority population.

## Figures and Tables

**Figure 1 ijerph-21-00282-f001:**
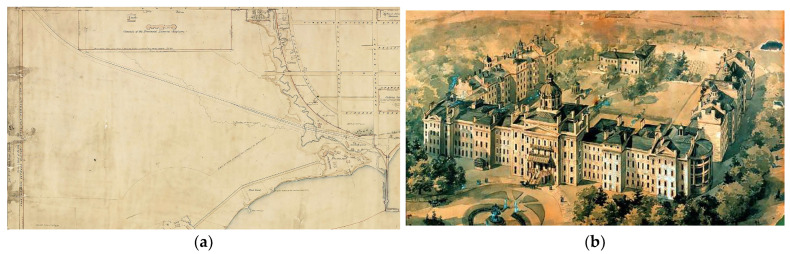
Pictured is the Queen Street campus location (historic). (**a**) The Provincial Lunatic Asylum, the first iteration of CAMH, sits on lands described as the council grounds and camping site of the Mississaugas of the Credit First Nation. (**b**) Pictured is the original structure of CAMH located on Queen Street (1850).

**Figure 2 ijerph-21-00282-f002:**
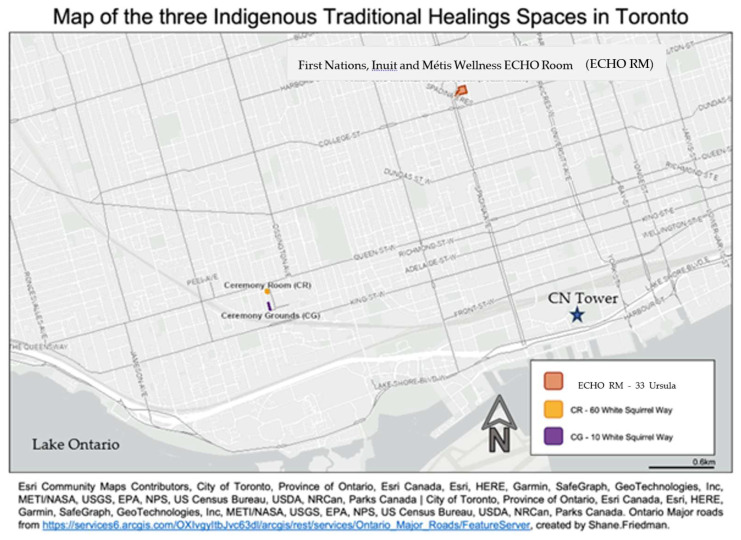
Map of CAMH’s THSs within the City of Toronto. The THSs are centrally located in the downtown City of Toronto, with the CR and CG located on White Squirrel Way, within CAMH’s main campus on Queen Street West. The third space, the ECHO RM, is located several blocks north, at the 33 Ursula Franklin Way site. At the time of publication, the ECHO RM had moved from this location to the main campus, on the second floor of 60 White Squirrel Way (2022).

**Figure 3 ijerph-21-00282-f003:**
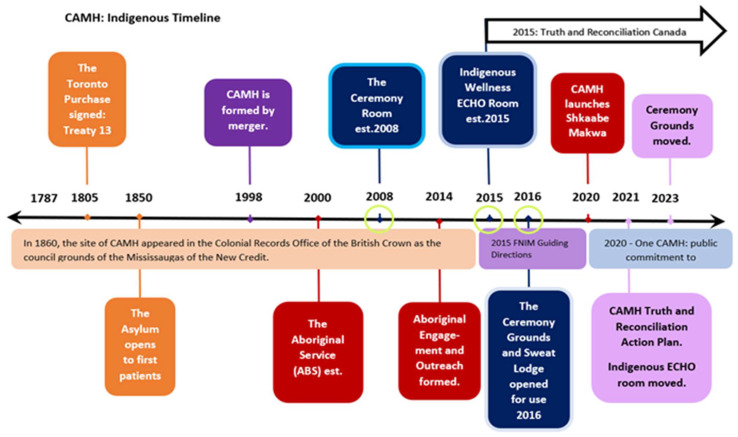
Pictured is a timeline (not to scale) of distinct and important events related to CAMH advancing spaces and programs for Indigenous clients, patients, and staff. Documentation about the evolution and development of the THSs at CAMH is limited; however, timelines were indicated through reports and formal public communiques on CAMH’s website and through participants’ stories. (This timeline offers a bird’s eye view of the evolution of Indigenous spaces within CAMH and serves as an important contextual point of reference.) Green circles have been placed around three years in the timeline, signifying the year that each of the three THS was established.

**Figure 4 ijerph-21-00282-f004:**
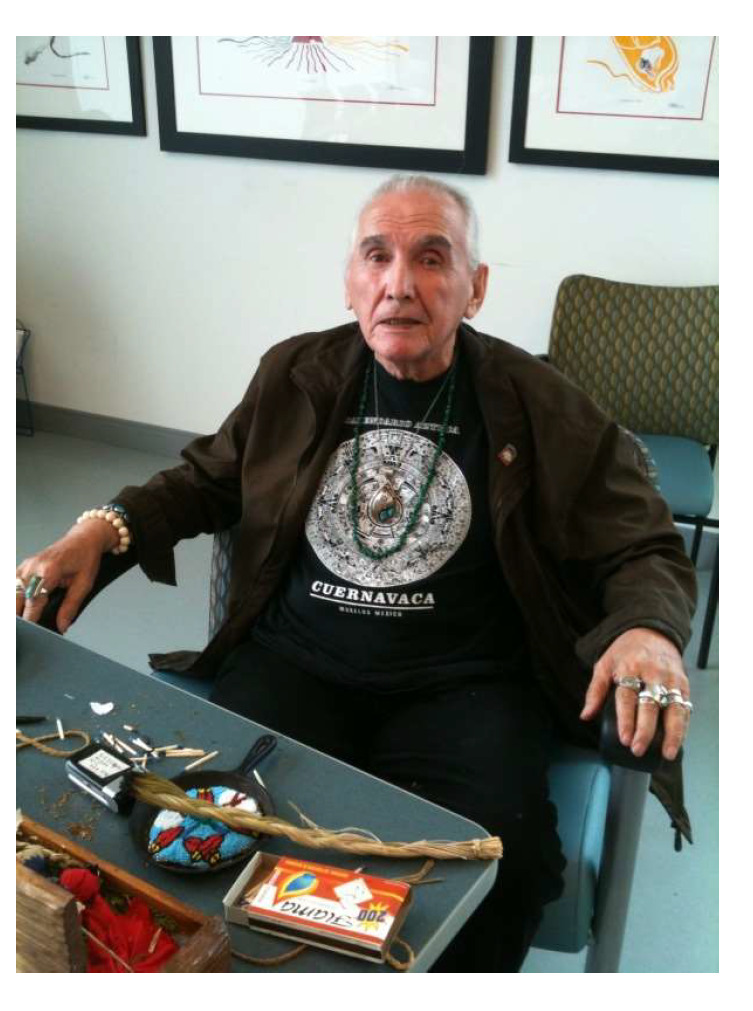
Asin Vern Harper baa (1936–2018) pictured in 2011, during his time at CAMH. Image courtesy of R.L. (5th author).

**Figure 5 ijerph-21-00282-f005:**
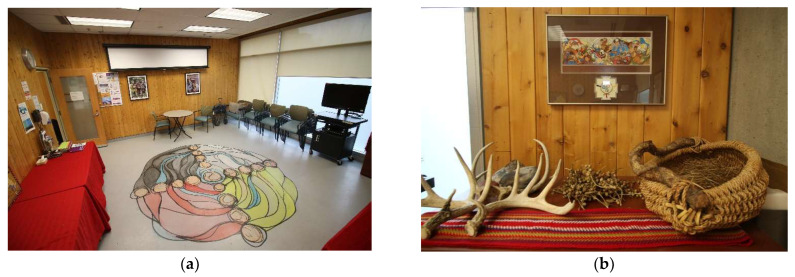
The Ceremony Room (CR) at CAMH. Pictured is the Ceremony Room, located at 60 White Squirrel Way, on the main campus on Queen Street West in Toronto, Ontario. (**a**) Indigenous artists transformed this space by painting the medicine wheel mural in the centre of the floor as pictured (2013). Tables were removed, and chairs are often placed around the mural to support talking circles. (**b**) The CR features walls lined with cedar planks and house Indigenous artwork; the only tables used are for traditional medicines/objects for ceremonial use.

**Figure 6 ijerph-21-00282-f006:**
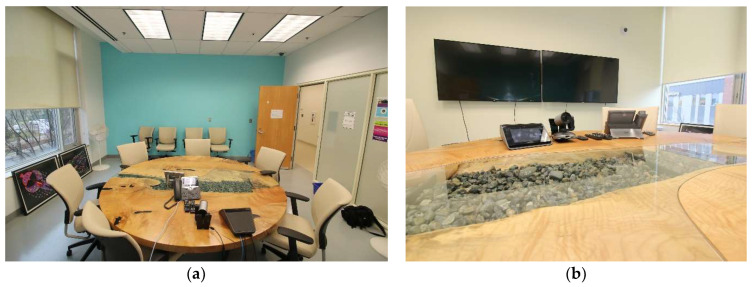
Pictured is the current room at CAMH, once it was relocated to the main campus at 60 White Squirrel Way in 2021. (**a**) The custom blue-painted wall serves a focal point for virtual participants and is adorned with Indigenous artwork. (**b**) The central feature of the room is a custom round table that ties in natural elements into the space, such as wood, stones, and water.

**Figure 7 ijerph-21-00282-f007:**
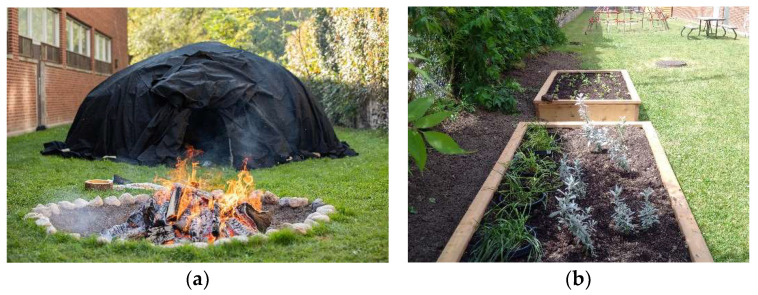
Pictured are the Ceremony Grounds at CAMH, one of the THSs. (**a**) The Sweatlodge ready for ceremony and covered with a black canvas, located south of the Sacred Fire pit, used to heat the stones used in the ceremonial lodge. The oblong shape of the Sweatlodge is intentional, “The lodge represents the womb of our mother, Mother Earth, as our first mother” (White 2021, 5:00). (**b**) The medicine gardens, used to grow plants (sacred medicines) such as traditional sacred tobacco, cedar, sage, and sweetgrass, used as part of these Traditional Healing ceremonies.

**Figure 8 ijerph-21-00282-f008:**
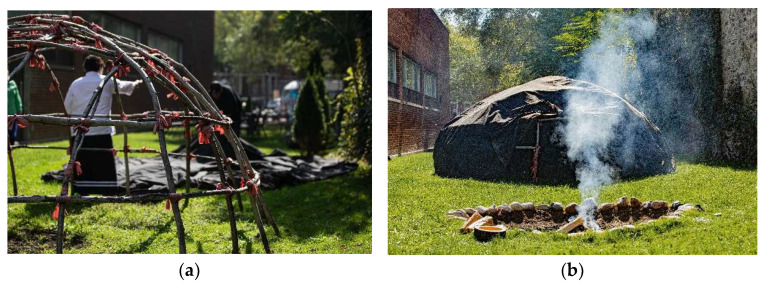
Pictured is the Sweatlodge ceremony preparations. (**a**) The bare willow tree frame of the Sweatlodge as it is being prepared by participants for ceremony. (**b**) The Sweatlodge covered with black canvas located adjacent to the Sacred Fire, where the stones are heated for the ceremony. Visible to the right of the Sweatlodge is the original stone wall from the 1850s asylum.

**Table 1 ijerph-21-00282-t001:** Characteristics of interviewees (*n* = 22).

Self-Identifies as Indigenous ^a^ (y/n)	Gender ^b^	Hospital Role ^c^
Y: 64% (*n* = 14) Self-identified as Indigenous, meaning either First Nations, Inuit, or Métis.	77% (*n* = 17) Identified as female	60% (*n* = 13) Formal leadership role (directors and managers)
N: 36% (*n* = 8) Self-identified as a non-Indigenous person	18% (*n* = 4) Identified as male	60% (*n* = 13) Located within an Indigenous team
No response: 0	5% (*n* = 1) Identified as non-binary	23% (*n* = 5) Clinicians (including leaders, Indigenous teams)
-	-	18% (*n* = 4) Other

^a^ Do you self-identify as an Indigenous (First Nations, Inuit, or Métis) person, Y or N? ^b^ What gender do you identify with? ^c^ What is your role at CAMH? They can be co-listed in more than one category.

**Table 2 ijerph-21-00282-t002:** When were the THS established and what was its original purpose?

Traditional Healing Spaces (THSs)	Original Purpose of the THSs
The Ceremony Room (est. in 2008)	The original purpose of the first THS at CAMH, the Ceremony Room, was for Indigenous clients of the ABS.
The Ceremony Grounds with medicines gardens, a Sweatlodge and Sacred Fire (est. in 2016)	Thus, the original intent and purpose of the Ceremony Grounds was to house a Sweatlodge for the use of Indigenous clients’ healing and mental health treatment.
The First Nations, Inuit, and Métis ECHO Room (est. in 2015)	Designed to support the relationships between a centralized interprofessional Healthcare Resource Team (located at CAMH, Toronto) to Community Providers, located in First Nations, Inuit, and Métis communities across Ontario.

**Table 3 ijerph-21-00282-t003:** Names used to identify the Traditional Healing Spaces (THSs).

Traditional Healing Spaces (THSs)	Number of Unique Names	Different Names Used to Refer to the Spaces
The Ceremony Room (est. 2008)	14	Ceremony Room; Room 116; 60 White Squirrel Way; “60”; Treatment Room; Healing Room; Ceremony Space; Cultural Room; Room 110; The Aboriginal Services Room; The smudge room; A sacred space; Cultural space; Medicine Room.
The Ceremony Grounds with medicine gardens, a Sweatlodge and Sacred Fire (est. 2016)	10	The Sacred Fire and Sweatlodge; The Sweatlodge; The Ceremony Grounds; The Sacred Fire; The Sweat Grounds; The Sacred Grounds (behind 10 White Squirrel Way); The Grounds; The Sacred Space; Sacred space out in the rear of 10 White Squirrel Way; The Sacred Space behind 10 White Squirrel Way;
The First Nations, Inuit and Métis Wellness ECHO Room (est. 2015)	8	Echo Room; Ursula Franklin; The ECHO Indigenous Room; FNIM ECHO Room; Telehealth Room; 33 Russell Street; First Nations, Inuit, and Metis ECHO Ontario Wellness Project; The First Nations, Inuit, and Métis Virtual Tele-mental Health Room

Table 3 illustrates an interesting finding: there were a total of 32 different names used to refer to the 3 THSs during the interviews, with no indication of a formally adopted name that was used consistently.

**Table 4 ijerph-21-00282-t004:** What Traditional Healing Spaces (THSs) were accessed the most by study participants?

Traditional Healing Spaces (THS)	Space Accessed by Study Participants (y/n)
The Ceremony Room (established in 2008)	Yes: 91% (*n* = 20)
	No: 9% (*n* = 2)
The Ceremony Grounds with medicines gardens, a Sweatlodge and Sacred Fire (established in 2016)	Yes: 86% (*n* = 19)
	No: 14% (*n* = 3)
The First Nations, Inuit, and Métis ECHO Room (established in 2015)	Yes: 50% (*n* = 11)
	No: 50% (*n* = 11)

## Data Availability

Data from this study are not available publicly due to ethical agreements to ensure that the data are secured and stewarded by Indigenous-led protocols around access to research involving First Nations, Inuit, and Metis Peoples in Canada (Chapter 9, Tri-Council Policy Statement on the Ethical Conduct for Research Involving Humans—TCPS 2, retrieved from: https://ethics.gc.ca/eng/tcps2-eptc2_2022_chapter9-chapitre9.html (accessed on 1 August 2020).

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
