# Peer review of "Incorporating First Nations, Inuit and Métis Traditional Healing Spaces within a Hospital Context: A Place-Based Study of Three Unique Spaces within Canada’s Oldest and Largest Mental Health Hospital"

_ijerph, 2024, doi:10.3390/ijerph21030282_

Round 1

Reviewer 1 Report

Comments and Suggestions for Authors

Hello and thank you for the opportunity to review this important article. I provide a list of comments and edits up to line 188 for your consideration. I feel that I am getting into full on editing, which I think is needed to better represent your important research and associated findings. 

·      Recommend using the following capitalization throughout, because as written, it is inconsistently capitalized: Indigenous Peoples

·      Serial comma usage is inconsistent throughout

·      Many sentences start the word this, which is not the strongest of writing and it is difficult to read

·      Line 56: Canada was colonized by the British and the French

·      Line 59: the quote is very long so I am wondering if it should be a block text

·      Line 82: suggest adding the specific Calls to Action that is referenced in the sentence

·      Line 88: an extra s is in the sentence

·      The use of semi colons and colons is quite abundant and could, perhaps, be decreased improve the readability 

·      Line 101 – “that identity formed and a deep” (I think the word is might be missing in this part of the sentence)

·      Line 111: why are single quotations used around the word ‘Land’?

·      Line 113: I have not read quotations as standalone sentences before. Usually, authors write a short preamble to introduce the quote and give it context, something like:  As Lawford (year) described, “aslkdfa sdflasd…”

·      Line 132: “However, it was not for several decades before these…” Do you mean that it took several decades for Indigenous Peoples to exercise their rights to incorporate traditional practices for their health and wellness? As currently written, it reads that Indigenous Peoples practiced their customary health and wellness ways of life before it became legal to do so. Can you clarify the timing in relation to the 1951 amendments? 

·      Line 134: Recommend using the full phrase: Indian Residential School system

·      Line 135: The IRS was also about civilizing, not just assimilating

·      Line 148: please double check the correct subject-verb agreement: reflect or reflects

·      Line 148: this sentence is a bit confusing. Can you separate out the ideas to write two or three sentences? 

·      Line 157: why are the single quotations used? Is this the same or different as double quotations? Is this a direct quote? 

·      Line 159: What are the four realms? 

·      Line 157-163: I don’t understand the connection with the three sentences. Yes, they are all true but how are they related to each other? 

·      Line 170: Recommend capitalizing Calls to Action because they are formally described in the TRC

·      Line 174: The use of single quotes is confusing. Is this to denote a quotation? Sarcasm? Emphasis? Can you review throughout?

·      Line 185: Not all Indigenous Peoples on Turtle Island smudge. Certainly, smudging is a good example, but it is not a practice common to all Nations. For example, Inuit do not smudge. 

·      Line 188: can you confirm the text as written? It seems there are a few words missing or there are extra words

Comments on the Quality of English Language

See above

Author Response

To Reviewer #1 of Incorporating First Nations, Inuit and Métis traditional healing spaces within a hospital context: A place-based study of three unique spaces within Canada’s oldest and largest mental health hospital, I would like to thank you for taking the time to review this manuscript. Please find the detailed responses to each point, numbered below with my responses detailed in red font. I have incorporated the corresponding revisions/corrections, highlighted in yellow, in the re-submitted file entitled “ijerph-2829403_revisions”.

The peer review process can be challenging, and we greatly appreciate the quick turnaround and the time taken to detail the points listed below. I have similarly taken the time to document and detail our responses to every point listed below. I have noticed that in doing this revision process, there has been a great improvement in the clarity, organization and overall presentation of the results and research in this manuscript. I believe that by embracing the detailed points listed below, I have been able to incorporate changes resulting in a paper that reads better, and with increased accuracy. I want to say miigwetch and thank you to each of the reviewers and to the editorial team for your thoughtful recommendations and observations. Sincerely, Vanessa Ambtman-Smith, on behalf of the co-authors.

Point 1: Recommend using the following capitalization throughout, because as written, it is inconsistently capitalized: Indigenous Peoples

Response 1: I agree that there were many inconsistencies. I did a document search for all instances of “Indigenous People” and corrected so that there was capitalization in both words throughout.

Additional clarifications

[All instances where I edited the within the manuscript were highlighted in yellow; however, there are situations here I added in a reference, or capitalization, and the whole sentence is highlighted because I edited the sentence itself to reflect a change. An example of this is line 132, with the following citation “[1,13-14,56].” The addition of ‘56’ as a new reference source was added because of the recommendation from reviewer #3].

Point 2: Serial comma usage is inconsistent throughout.

Response 2: I agree. I ran through the document edit reviewer, and focused on the punctuation conventions and edited all the inconsistencies in serial comma use – the most common omission of comma use came after “Inuit” when listing amongst Indigenous populations.

Additional clarifications

[I went through and added commas in many lists in the document and noted with a yellow highlight. An example can be viewed in Table1, column 1, row 2 after the word “Inuit”.]

Point 3:  Many sentences start the word this, which is not the strongest of writing and it is difficult to read.

Response 3: Thank you for pointing this out. I did a document search for all sentences that started with “This” and was able to identify 30 sentences that started out this way. I chose to edit 11 of these sentences, where I saw that changing the sentence would reduce the use of a semi-colon, or where a change would strengthen the sentence as a whole. The changes I have made have greatly strengthened the manuscript. I made the following changes:

  • Lines 82 – 86 read: “This research responds in important ways to the 2015 release of the Truth and Reconciliation Report, which identified TH as a key healing practice among Indigenous Peoples and challenged Canadian hospitals to create and offer THS within their mainstream practice [12].” In keeping with Point 3 and Point 6, the sentence was edited to be: [Building on the importance of the Truth and Reconciliation Report (2015) Call to Action #22, this research responds to an evolving healthcare landscape as hospitals transform spaces to accommodate and “recognize the value of Aboriginal healing practices and use them in the treatment of Aboriginal patients in collaboration with Aboriginal healers and Elders where requested by Aboriginal patients”[12:p.3].]
  • Lines 381-383 read: “This work advanced as Indigenous-specific roles were developed at CAMH under the Aboriginal Services.” And was edited to read [The growth of THS was linked to the growth of the Aboriginal Services, with the launch of many more Indigenous-specific roles at CAMH].
  • Line 403 read: “This table illustrates an…” and was edited to read: [Table 3 illustrates an…].
  • Line 604 read “This is important as this finding reinforces the importance of …” and was changed to read [These findings reinforce the importance of relationship with Indigenous staff as key knowledge…].
  • Lines 677-682 formerly read: “This case study seeks to provide analysis on the phenomena of TH experiences within a mainstream healthcare environment that will contribute to an area of medicine that is not well understood.” It has been edited to read: [As a case study, this research provides preliminary analysis on the phenomena of THS within a mainstream healthcare environment, seeking to build knowledge in an area of scholarship that is still emerging. Findings indicate that while staff access THS to recover from colonial violence and trauma, there are other important uses of space for staff, including opportunities to access ethical spaces of engagements that are important for advancing Indigenous cultural safety and humility education.]
  • Lines 686-688 read: “While education was a regularly cited reason for using the THS, this may add additional burden to already overextended resources. This suggests that additional Indigenous spaces may be required to adequately facilitate reconciliation and decolonizing education of staff and visitors.” This was edited to read: [While education was a regularly cited reason for using the THS, this may overburden resources designed for patient use, suggesting a need for additional spaces for decolonizing and Indigenous education.]
  • Lines 709-710 was edited from “This distinction matters…” to “Distinction related to designation matters…”
  • Lines 719-721 read: “This may allude to a larger issue of access it was identified that change happens slowly within CAMH and its systems as an institution “it’s difficult to make those changes and raise awareness…it’s something that has always been a struggle, but slowly they have been able to make and modify the rooms” [Staff 1:p.4].” and was edited to read: [Staff identified that “it’s difficult to make those changes and raise awareness…it’s something that has always been a struggle, but slowly [CAMH has] been able to make and modify the rooms” [Staff 1:p.4], which may allude to a larger issue around access.]
  • Line 739 was changed from “This study on THS grows from…” to [The THS study grows from the geographies of Indigenous health and pulls from…].
  • Lines 765-766, where I removed a semi-colon (based on point 8) and changed the start of the sentence to remove the word “This” at the beginning: [A result of this increase are limits on what these spaces can and cannot support where significant growth has meant that there are needs that are not being met through the current spaces].
  • Lines 770-782, I edited the following sentence “This has specifically been called out by Ermine [49], wherein the time and place of reconciliation requires a new space for dialogue: this space creates a contrast by dislocating and isolating two disparate knowledge systems and cultures.” To read: [Creating a new space for dialogue has been called out by Ermine in 2009, who notes that these spaces create a contrast by dislocating and isolating two disparate knowledge systems and cultures [49]. Ermine emphasizes that ethical spaces of inquiry are required to build common ground between worldviews, noting “there have been lots of good attempts by sincere people who have tried to build bridges, but these undercurrents are powerful and keep washing away good intentions ... when we have had breaches and ruptures in the past, it is because we have failed to look at the area in between our two worlds” [49]. Tobias et al. (2013) remind us that Indigenous “knowledge systems are still greatly being applied to help Indigenous communities and Indigenous Peoples recover from intergenerational pain…future policy development and implementation should aim to support Indigenous Peoples and communities when they decide to learn about, maintain, and build upon the knowledge amassed by their ancestors” [50:p.3].]

Point 4: Line 56: Canada was colonized by the British and the French

Response 4: This is an important detail that can’t be ignored. I have therefore edited the sentence “…long before Canada was colonized as a country under the British monarchy” to read [long before Canada was colonized by the British and the French] line 56.

Point 5: Line 59: the quote is very long so I am wondering if it should be a block text?

Response 5: I agree that this quote is long, and after review, decided that it was not necessary to include the complete quote as recorded, so I edited it down to be shorter, so block text was not needed. The original quote and sentence was: This view describes this “Indigenous perspective [as] introspective, inward, subjective, and relational in its seeking to live with and to be in harmony and balance. Indigenous wholistic wellness and healing methodologies intentionally engage meaning and relationality in teaching, learning, and healing processes; engaging the spiritual, emotional, mental, and physical as a way of coming to see, to relate, to know and activate that relationality in the living of life” [8: p.253]. I edited this down to be: [A wholistic view is described as “introspective, inward, subjective, and relational in its seeking to live with and to be in harmony and balance. Indigenous wholistic wellness and healing methodologies intentionally engage meaning and relationality in teaching, learning, and healing processes; engaging the spiritual, emotional, mental, and physical as a way of coming to see, to relate, to know and activate that relationality in the living of life” [8: p.253].]

Point 6:  Line 82: suggest adding the specific Calls to Action that is referenced in the sentence.

Response 6: The text was edited to respond to both points 3 and 6, now reading: [Building on the importance of the Truth and Reconciliation Report (2015) Call to Action #22, this research responds to an evolving healthcare landscape as hospitals transform spaces to accommodate and “recognize the value of Aboriginal healing practices and use them in the treatment of Aboriginal patients in collaboration with Aboriginal healers and Elders where requested by Aboriginal patients”[12:p.3].]

Point 7: Line 88: an extra ‘s’ is in the sentence.

Response 7: Line 90: extra ‘s’ was erased.

Point 8: The use of semi colons and colons is quite abundant and could, perhaps, be decreased to improve the readability.

Response 8: I reviewed sentences with semi-colons and edited select ones that were high impact, e.g. at the start of a paragraph or section, and the following edits were made:

  • Lines 130-133 read: “A body of Indigenous research demonstrates the profound health and cultural impacts experienced by Indigenous Peoples because of ongoing processes of colonialism and land dispossession [1,13-14,56]; this body of work argues that cultural wounds require cultural medicine, begging the importance of introducing Indigenous healing practices and spaces into biomedical environments [3].” I changed this sentence to: [A growing body of Indigenous research connects processes of colonialism and land dispossession to profound health and cultural impacts, arguing that cultural wounds require cultural medicine[1,13-14,56]. Therefore, we recognize the importance of introducing Indigenous healing practices and spaces into biomedical environments [3].]
  • I edited the following sentence from “There have been lots of good attempts by sincere people who have tried to build bridges, but these undercurrents are powerful and keep washing away good intentions ... When we have had breaches and ruptures in the past, it is because we have failed to look at the area in between our two worlds” [49]; therefore, an ‘ethical space’ of inquiry and understanding is required to acknowledge and close the gap of difference in worldview around health and healing through one another's knowledge systems and worldview.” To remove semicolon and reframe to [Ermine emphasizes that ethical spaces of inquiry are required to build common ground between worldviews, noting “there have been lots of good attempts by sincere people who have tried to build bridges, but these undercurrents are powerful and keep washing away good intentions ... when we have had breaches and ruptures in the past, it is because we have failed to look at the area in between our two worlds” [49].]
  • Based on point 23 in lines 196-198, the sentence needed to be edited to make sense and a semi-colon was removed, resulting in the following: [The process of decolonization has been defined by Corntassel as “embracing a daily existence conditioned by place-based cultural practices” [37: p.89] and “involves unlearning… and embracing messiness and tensions in the process of transformation” [38: p.17].] in lines 196-198.

Point 9: Line 101 – “that identity formed and a deep” (I think the word is might be missing in this part of the sentence)

Response 9: There were two words missing (“to place”) from this phrase, and it was corrected to read: [It is through a longstanding connection to place that identity formed and a…] on line 102.

Point 10: Line 111: why are single quotations used around the word ‘Land’?

Response 10: I agree that the use of single quotations in this context are unnecessary and could be confusing. Therefore, I omitted them from two instances in this paragraph, lines 112 and 114.

Point 11: Line 113: I have not read quotations as standalone sentences before. Usually, authors write a short preamble to introduce the quote and give it context, something like:  As Lawford (year) described, “aslkdfa sdflasd…”

Response 11: I noticed that there were 2 instances of standalone quotes as sentences while I was reviewing and editing this paper based on reviewer suggestions. I have been able to correct this practice and incorporate an appropriate preamble in each case as follows:

  • Lines 115-118 were edited first (based on point 11) to read: [In 2020, Allen et al. emphasize that “perhaps most importantly, traditional knowledge and Indigenous spirituality hinges on the maintenance and renewal of relationships to the land. Indigenous land bases and the environment as a whole remain vitally important to the practice of traditional healing” [6:p.180].]
  • Edited lines 770-782 that read “Indigenous Knowledge systems are living entities and not relics of the past. Today, these knowledge systems are still greatly being applied to help Indigenous communities and Indigenous Peoples recover from intergenerational pain and suffering endured during the colonization process. Future policy development and implementation should aim to support Indigenous Peoples and communities when they decide to learn about, maintain, and build upon the knowledge amassed by their ancestors” [50:p.3]. I changed this to ensure that the quote was shorter and not a stand-alone sentence: [Creating a new space for dialogue has been called out by Ermine in 2009, who notes that these spaces create a contrast by dislocating and isolating two disparate knowledge systems and cultures [49]. Ermine emphasizes that ethical spaces of inquiry are required to build common ground between worldviews, noting “there have been lots of good attempts by sincere people who have tried to build bridges, but these undercurrents are powerful and keep washing away good intentions ... when we have had breaches and ruptures in the past, it is because we have failed to look at the area in between our two worlds” [49]. Tobias et al. (2013) remind us that Indigenous “knowledge systems are still greatly being applied to help Indigenous communities and Indigenous Peoples recover from intergenerational pain…future policy development and implementation should aim to support Indigenous Peoples and communities when they decide to learn about, maintain, and build upon the knowledge amassed by their ancestors” [50:p.3].]

Point 12: Line 132: “However, it was not for several decades before these…” Do you mean that it took several decades for Indigenous Peoples to exercise their rights to incorporate traditional practices for their health and wellness? As currently written, it reads that Indigenous Peoples practiced their customary health and wellness ways of life before it became legal to do so. Can you clarify the timing in relation to the 1951 amendments? 

Response 12: In lines 135-143, the sentence leading up to this and the actual identified sentence were both overly wordy and confusing, so I edited them to read [Without access to the healing modalities supported through TH and ceremonial practices that were banned under the Indian Act until 1951, mental illness and the negative impacts of trauma proliferated amongst Indigenous populations [4]. Given the severity of punishment and the residual fear associated with openly practicing ceremony, compounded by loss of traditional knowledge at the community level, access to TH remained limited for many decades following the 1951 legislative amendments. The resurgence of traditional knowledge systems and the implementation of TH practice in public spaces, especially colonial institutions, is a relatively new and growing practice over the 30 years [25,35-36].

Point 13: Line 134: Recommend using the full phrase: Indian Residential School system.

Response 13: The partial phrase of “residential schools” was expanded to use the full phrase “the Indian Residential School System” on line 143.

Point 14: Line 135: The IRS was also about civilizing, not just assimilating.

Response 14: I agree and added the term “civilizing” to the phrase, which now reads: [Since the time of the Indian Residential School System, institutional spaces have been widely characterized as a tool in assimilating and civilizing the original people of Turtle Island…] on lines 143-145.

Point 15 and 16: Line 148: please double check the correct subject-verb agreement: “reflect” or “reflects”; and this sentence is a bit confusing. Can you separate out the ideas to write two or three sentences?

Response 15 and 16: Doing a careful review of lines 147-155 led me to catch a few issues with grammar, and I ended up editing a few details from the original here: “Despite this negative history, institutional spaces continue to be used widely as purveyors of healthcare, education, and justice [26]. Specifically, hospitals have been called out as unsafe spaces for Indigenous Peoples [10-11, 22]. Yet, within Canadian healthcare systems, hospitals are a foundational space for the delivery of healthcare services, and are built on a predominantly, western, bio-medical system. As a result, Indigenous Peoples continued to experience limited care within these institutions, and in fact, important determinants of Indigenous health, such as physical connections to cultural continuity and connections with the natural environment, were not possible through a western, bio-medical model in hospital.”  This paragraph was edited to be: [Despite this negative history, institutional spaces continue to be used widely as purveyors of healthcare, education, and justice [26]. Specifically, hospitals have been called out as unsafe spaces for Indigenous Peoples [10-11, 22]. Yet, within Canadian healthcare systems, hospitals are a foundational space for the delivery of healthcare services, and built predominantly on a western, bio-medical system that excludes Indigenous TH and access to the land. Thus, important determinants of Indigenous health, such as cultural continuity and connections with the natural environment were largely absent or ignored in those places, limiting access to the wholistic healthcare practices needed to improve Indigenous Peoples health outcomes.] on lines 146-154.

Point 17,18 and 19: Line 157: why are the single quotations used? Is this the same or different as double quotations? Is this a direct quote? Line 159: What are the four realms? Line 157-163: I don’t understand the connection with the three sentences. Yes, they are all true but how are they related to each other? 

Response 17,18 and 19: I did a review of this section and can see that there were a few confusing aspects to the context and phrasing. Therefore, I edited and made significant gains in clarification of lines 159-172, which now reads: [Despite advances in modern healthcare, including increased social services, dedicated resources for research and the implementation of abundant Indigenous health programs, very little improvement in the overall health and wellbeing of Indigenous Peoples has been documented. The limits of improvement point to ongoing structural and systemic disparity wherein the current systems are not adequately equipped to attend to these inequalities [1, 3, 4, 53]. Disconnections to culture through dispossession has been linked with a diminished cultural identity, and poor mental health and wellbeing [21; 32-33; 57]. Outcomes include a sickness that begins within the spirit: spiritual and cultural imbalances manifest through interconnections in health, permeating sickness through to the mind, emotions, and body [7; 33-34; 57].] Additionally, edits were made to describe the four realms in lines 170 and 171 to read: [True healing includes re-establishing a balance between the four realms of the physical, mental, emotional and spiritual [34] wherein…].

Point 20: Line 170: Recommend capitalizing Calls to Action because they are formally described in the TRC.

Response 20: I have corrected this oversight to be consistent with line 181 in capitalizing Calls to Action on line 183.

Point 21: Line 174: The use of single quotes is confusing. Is this to denote a quotation? Sarcasm? Emphasis? Can you review throughout?

Response 21: I reviewed the use and meaning of the single quotes used in lines 187-188 and determined that the quotes appeared around three terms (colonization, colonialism, and decolonizing) that I was defining as core concepts. The rationale for using these single quotes in identifying concepts was not explained, nor is it consistent throughout the paper, therefore it has been removed from all three words (colonization, colonialism, and decolonizing).

Point 22: Line 185: Not all Indigenous Peoples on Turtle Island smudge. Certainly, smudging is a good example, but it is not a practice common to all Nations. For example, Inuit do not smudge. 

Response 22: This text was reviewed and given the wording of the sentence that alludes to a widespread practice of smudging, edits were made to the sentence, which originally read: “To offer traditional healing practices effectively, space must be transformed, or created to enable access to a land-base, or to use land-based medicines for smudging [26,30-32].” The edited sentence now reads [To offer TH practices effectively, institutional spaces must be transformed to facilitate access to or connections with the natural environment, which may include the burning of plant medicines, often referred to as smudging [26,30-32].] and is found on lines 197-199.

Point 23: Line 188: can you confirm the text as written? It seems there are a few words missing or there are extra words

Response 23: Upon review, the sentence in lines 188-191 needed to be edited to make sense and was edited to the following: [The process of decolonization has been defined by Corntassel as “embracing a daily existence conditioned by place-based cultural practices” [37: p.89] and “involves unlearning… and embracing messiness and tensions in the process of transformation” [38: p.17].]

Reviewer 2 Report

Comments and Suggestions for Authors

In the intro, you say "This paper describes..." - personal issue - papers are inanimate objects. You or Our team describes - papers don't.

You were using TH for traditional healing and then started to use THS. What is the difference? It was not explained.

You switch between SweatLodge and Sweat Lodge. Please pick one.

First line of conclusion: "The research amplifies to need..." - maybe "Our findings amplify a need for..."

Can you add a positionality statement in the paper?

Great paper!! 

Comments on the Quality of English Language

Check the acronyms and a couple of lines are awkward (as shown in the suggestions).

Author Response

Dear Reviewer #2:

I would like to thank you for taking the time to review this manuscript. Please find the detailed responses to each point, numbered below with my responses detailed in red font. I have incorporated the corresponding revisions/corrections, highlighted in yellow, in the re-submitted file entitled “ijerph-2829403_revisions”.

The peer review process can be challenging, and we greatly appreciate the quick turnaround and the time taken to detail the points listed below. I have similarly taken the time to document and detail our responses to every point listed below. I have noticed that in doing this revision process, there has been a great improvement in the clarity, organization and overall presentation of the results and research in this manuscript. I believe that by embracing the detailed points listed below, I have been able to incorporate changes resulting in a paper that reads better, and with increased accuracy. I want to say miigwetch and thank you to each of the reviewers and to the editorial team for your thoughtful recommendations and observations. 

Point 1: In the intro, you say "This paper describes..." - personal issue - papers are inanimate objects. You or Our team describes - papers don't.

Response 1: I agree with your points, and have found that by implementing these changes, the clarity of my paper has improved. For example, in my introduction, I implemented your suggestion, and the resulting sentence in lines 23-25 now reads: [In this paper, our team describes THS within the Center for Addiction and Mental Health (CAMH), Canada’s oldest and largest mental health hospital.]

Additional clarifications

[I needed to omit the word “being” in this sentence, and this is noted in line 24 by a yellow highlight on the comma after CAMH, and before “Canada”, where this word was removed.]

Point 2: You were using TH for traditional healing and then started to use THS. What is the difference? It was not explained.

Response 2: This is a keen observation! I did document traditional healing as TH on line 46, but did not document what THS were when it was first used on line 78. I have now edited the sentence in lines 78-80 to read: [This paper describes the creation and use of traditional healing spaces (THS) at the Centre for Addiction and Mental Health (CAMH), Canada’s oldest and largest mental health hospital, located in Toronto, Ontario, Canada.]

Point 3: You switch between Sweatlodge and Sweat Lodge. Please pick one.

Response 3: My co-authors recommend using “sweatlodge”, and I have done a document search to locate any instances of “sweat lodge”. 7 occurrences were found, and I edited each one to be consistent with the rest of the document by using “sweatlodge”.

Additional clarifications

[The 7 occurrences of “sweat lodge” as two words instead of the recommended “sweatlodge” denoted as one word are highlighted in each instance, using yellow highlighter in text.]

Point 4: First line of conclusion: "The research amplifies to need..." - maybe "Our findings amplify a need for..."

Response 4: This is a grammatical error that impacts the clarity of the sentence. I have taken your suggestion and made the edits on line 758 to read [Our findings amplify a need for a geographical lens in Indigenous health research and seeks to serve as a promising example of institutional transformation for any healthcare, education, or other institution involved in Indigenization and reconciliation efforts [26].]

Point 5: Can you add a positionality statement in the paper?

Response 5: I agree that given the nature of this scholarship, a positionality statement is important, so we added one in lines 259-264: [This research is part of the PhD project of the first author, who is of mixed ancestry (Cree and Métis) and has worked in the Indigenous health field for the past two decades. During her tenure, she collaborated extensively with health organizations, including CAMH, to advance Indigenous health equity. The last author is an Anishinabe scholar from Biigtigong Nishnaabeg and is the academic supervisor to the first author].

Point 6: Check the acronyms and a couple of lines are awkward (as shown in the suggestions).

Response 6: Significant editing throughout the manuscript to clarify has taken place based on your comments and reviewers 1 and 3, including a review of the acronyms. Thank you for the notes!

Reviewer 3 Report

Comments and Suggestions for Authors

Thank you for the privilege or reviewing your well written and timely paper about the vital importance of inclusion of First Nations traditional healing spaces within hospitals and mainstream health settings.  This is a really important paper that poses some important questions for consideration by mainstream health services and beyond in the conclusion.  I so enjoyed reading it and know it will make a valuable and important contribution to literature.

A several minor edits required (just typos) and I have made a few suggestions re inclusion of literature etc., mainly to internationalize this paper for a global audience, and also support all that you have discussed so well in your paper. II have also indicated with comments within the PDF of your paper where these comments apply.

I absolutely recommend this paper for publication

Abstract: Line 22, have should read has?

Introduction

Lines 44 and 45: As this is an international journal, and you are citing this as an international phenomenon, citing literature about parallel and similar experiences of  other Indigenous Peoples colonized by western invaders would be appropriate here:

For example:

Parter, C., Gwynn, J., Wilson, S., Skinner, J. C., Rix, E., & Hartz, D. (2024). Putting Indigenous Cultures and Indigenous Knowledges Front and Centre to Clinical Practice: Katherine Hospital Case Example. International Journal of Environmental Research and Public Health, 21(1), 3. https://www.mdpi.com/1660-4601/21/1/3

Asamoah, G. D., Khakpour, M., Carr, T., & Groot, G. (2023). Exploring Indigenous Traditional Healing programs in Canada, Australia, and New Zealand: A scoping review. EXPLORE, 19(1), 14-25. https://doi.org/https://doi.org/10.1016/j.explore.2022.06.004

NOTE: Ah I have just seen that you have cited this paper further down in your article, great we are on the same track here J

Around a ref here also to line 121, the crucial and relational connection to land is also a strong theme for all colonized Indigenous Peoples.  For example in Australia it is known as connection to Country, and is actually a key ingredient in all well-being and healing practices, without which not just the individual, but the entire community is impacted, See for example:

Terare, M., & Rawsthorne, M. (2019). Country Is Yarning to Me: Worldview, Health and Well-Being Amongst Australian First Nations People. The British Journal Of Social Work, 50(3). https://doi.org/10.1093/bjsw/bcz072

Lines 152 to 163:  

Line200-201: it may be worth mentioning briefly the Australian Aboriginal and Torres Strait Islander term and framework of Social and Emotional well-being and it's increasing inclusion in mainstream Mental health paradigms:

See for example:

Dudgeon, P., Milroy, H., & Walker, R. (Eds.). (2014). Working Together: Aboriginal and Torres Strait Islander Mental Health and Wellbeing Principles and Practice (2nd ed.). Commonwealth of Australia. https://www.telethonkids.org.au/globalassets/media/documents/aboriginal-health/working-together-second-edition/working-together-aboriginal-and-wellbeing-2014.pdf.

This textbook is freely available on-line.

Line220: sum, should this be summary?

Line 244: is this referencing numbering correct, as 43 and 44 are further down? just checking….

Line 270:  You may also want to consider using the term Case Example? to separate your methods and language from the biomedical terms and meaning of case study, only a suggestion here.

Lines 306-320:  This sounds rigorous and appropriate analysis here, where both worldviews are considered and included as ways of knowing being and doing research, thank you!

Figure 3: (after line 340): This Figure is really useful and provides the reader with more insight on the historical and still emerging exclusion/inclusion of Indigenous knowledges and highlights their exclusion until very recently within the  western and biomedical dominated  knowledges and practice.  This further validates your important research and methods.

Discussion

Line 653: grow should read as grown?

Line 675 to 678: This is a really important point you make here and agree I that non-Indigenous access to TH spaces etc can really help broaden understanding for clinicians and power holders within mainstream health. Reducing racism is a hard and ongoing challenge, an d any strategy that can increase positive   therapeutic relationships through a deeper connection and understanding needs prioritizing

Author Response

Dear Reviewer #3:

I would like to thank you for taking the time to review this manuscript. Please find the detailed responses to each point, numbered below with my responses detailed in red font. I have incorporated the corresponding revisions/corrections, highlighted in yellow, in the re-submitted file entitled “ijerph-2829403_revisions”.

The peer review process can be challenging, and we greatly appreciate the quick turnaround and the time taken to detail the points listed below. I have similarly taken the time to document and detail our responses to every point listed below. I have noticed that in doing this revision process, there has been a great improvement in the clarity, organization and overall presentation of the results and research in this manuscript. I believe that by embracing the detailed points listed below, I have been able to incorporate changes resulting in a paper that reads better, and with increased accuracy. I want to say miigwetch and thank you to each of the reviewers and to the editorial team for your thoughtful recommendations and observations. 

Point 1: Several minor edits required (just typos) and I have made a few suggestions re: inclusion of literature etc., mainly to internationalize this paper for a global audience, and also support all that you have discussed so well in your paper. I have also indicated with comments within the PDF of your paper where these comments apply. I absolutely recommend this paper for publication.

Response 1: I have copied and pasted each of your points, and provided a corresponding response in red. 

Point 2: Abstract: Line 22, ‘have’ should read ‘has’?

Response 2: This is a good edit, and I have made the change in line 22, which now reads as “has”.

Point 3: Lines 44 and 45: As this is an international journal, and you are citing this as an international phenomenon, citing literature about parallel and similar experiences of other Indigenous Peoples colonized by western invaders would be appropriate here. For example:

Parter, C., Gwynn, J., Wilson, S., Skinner, J. C., Rix, E., & Hartz, D. (2024). Putting Indigenous Cultures and Indigenous Knowledges Front and Centre to Clinical Practice: Katherine Hospital Case Example. International Journal of Environmental Research and Public Health, 21(1).

Response 4: I reviewed the source and agree that this helps to connect to the manuscript and broaden the global views contained therein. I added this source into the paper, and it appears as citation 55, and is found in line 46.

Point 4: Around a ref here also to line 121, the crucial and relational connection to land is also a strong theme for all colonized Indigenous Peoples.  For example, in Australia it is known as connection to Country, and is actually a key ingredient in all well-being and healing practices, without which not just the individual, but the entire community is impacted, See for example:

Terare, M., & Rawsthorne, M. (2019). Country Is Yarning to Me: Worldview, Health and Well-Being Amongst Australian First Nations People. The British Journal Of Social Work, 50(3).

Response 4: I reviewed the source and agree that this helps to connect to the manuscript and broaden the global views contained therein. I added this source into the paper, and it appears as citation 56, and is found in line 142.

Point 5: Lines 152 to 163 and Lines 200-201: it may be worth mentioning briefly the Australian Aboriginal and Torres Strait Islander term and framework of Social and Emotional well-being and it's increasing inclusion in mainstream Mental health paradigms. See for example:

Dudgeon, P., Milroy, H., & Walker, R. (Eds.). (2014). Working Together: Aboriginal and Torres Strait Islander Mental Health and Wellbeing Principles and Practice (2nd ed.). Commonwealth of Australia. 

Response 5: I added this source into the paper, and it appears as citation 57, in lines 166 and 213. In addition, I wanted to make specific reference to this framework, and included a new quote from this source, found on lines 192-194: [Detailed within the national framework for Aboriginal and Torres Strait Islander health, the inter-relationships to the land are identified as central to social and emotional wellbeing, wherein disruption to this connection results in ill health [57:p.xxiv].]

Point 6: Line220: sum, should this be summary?

Response 6: This indeed should be edited to say “summary” and I have done so on line 233.

Point 7: Line 244: is this referencing numbering correct, as 43 and 44 are further down? just checking….

Response 7: I double checked the numbering system, and given the format of the MDPI system, numerous edits and changes have resulted in the current numbering system and somewhat illogical order; however, it does line up to the references in text.

Point 8: Line 270:  You may also want to consider using the term Case Example? to separate your methods and language from the biomedical terms and meaning of case study, only a suggestion here.

Response 8: I took your suggestion and edited the term “case study” to read as “case example” in line 285.

Point 9: Lines 306-320:  This sounds rigorous and appropriate analysis here, where both worldviews are considered and included as ways of knowing being and doing research, thank you!

Point 10: Figure 3: (after line 340): This Figure is really useful and provides the reader with more insight on the historical and still emerging exclusion/inclusion of Indigenous knowledges and highlights their exclusion until very recently within the western and biomedical dominated knowledges and practice. This further validates your important research and methods.

Response to 9 and 10: I appreciate this!

Point 11: Line 653: grow should read as grown?

Response 11: Line 670 has been edited to read “grown” vs. “grow”.

Point 12: Line 675 to 678: This is a really important point you make here and agree I that non-Indigenous access to TH spaces etc. can really help broaden understanding for clinicians and power holders within mainstream health. Reducing racism is a hard and ongoing challenge, and any strategy that can increase positive therapeutic relationships through a deeper connection and understanding needs prioritizing.

Response to point 12: Miigwetch and thank you for your supportive comments!